# MAKING STOCHASTIC NEURAL NETWORKS FROM DETERMINISTIC ONES

**Kimin Lee,     Jaehyung Kim,     Song Chong,     Jinwoo Shin**

School of Electrical Engineering
Korea Advanced Institute of Science Technology, Republic of Korea

{kiminlee, jaehyungkim, jinwoos}@kaist.ac.kr, songchong@kaist.edu

## ABSTRACT

It has been believed that stochastic feedforward neural networks (SFNN) have several advantages beyond deterministic deep neural networks (DNN): they have more expressive power allowing multi-modal mappings and regularize better due to their stochastic nature. However, training SFNN is notoriously harder. In this paper, we aim at developing efficient training methods for large-scale SFNN, in particular using known architectures and pre-trained parameters of DNN. To this end, we propose a new intermediate stochastic model, called Simplified-SFNN, which can be built upon any baseline DNN and approximates certain SFNN by simplifying its upper latent units above stochastic ones. The main novelty of our approach is in establishing the connection between three models, i.e., DNN $\rightarrow$ Simplified-SFNN $\rightarrow$ SFNN, which naturally leads to an efficient training procedure of the stochastic models utilizing pre-trained parameters of DNN. Using several popular DNNs, we show how they can be effectively transferred to the corresponding stochastic models for both multi-modal and classification tasks on MNIST, TFD, CIFAR-10, CIFAR-100 and SVHN datasets. In particular, our stochastic model built from the wide residual network has 28 layers and 36 million parameters, where the former consistently outperforms the latter for the classification tasks on CIFAR-10 and CIFAR-100 due to its stochastic regularizing effect.

## 1 INTRODUCTION

Recently, deterministic deep neural networks (DNN) have demonstrated state-of-the-art performance on many supervised tasks, e.g., speech recognition (Hinton et al., 2012a) and object recognition (Krizhevsky et al., 2012). One of the main components underlying these successes is on the efficient training methods for deeper and wider DNNs, which include backpropagation (Rumelhart et al., 1988), stochastic gradient descent (Robbins & Monro, 1951), dropout/dropconnect (Hinton et al., 2012b; Wan et al., 2013), batch/weight normalization (Ioffe & Szegedy, 2015; Salimans & Kingma, 2016), and various activation functions (Nair & Hinton, 2010; Gulcehre et al., 2016). On the other hand, stochastic feedforward neural networks (SFNN) (Neal, 1990) having random latent units are often necessary in order to model complex stochastic natures in many real-world tasks, e.g., structured prediction (Tang & Salakhutdinov, 2013), image generation (Goodfellow et al., 2014) and memory networks (Zaremba & Sutskever, 2015). Furthermore, it has been believed that SFNN has several advantages beyond DNN (Raiko et al., 2014): it has more expressive power for multi-modal learning and regularizes better for large-scale learning.

Training large-scale SFNN is notoriously hard since backpropagation is not directly applicable. Certain stochastic neural networks using continuous random units are known to be trainable efficiently using backpropagation under the variational techniques and the reparameterization tricks (Kingma & Welling, 2013). On the other hand, training SFNN having discrete, i.e., binary or multi-modal, random units is more difficult since intractable probabilistic inference is involved requiring too many random samples. There have been several efforts developing efficient training methods for SFNN having binary random latent units (Neal, 1990; Saul et al., 1996; Tang & Salakhutdinov, 2013; Bengio et al., 2013; Raiko et al., 2014; Gu et al., 2015) (see Section 2.1 for more details). However, training SFNN is still significantly slower than doing DNN of the same architecture, e.g., most prior

works on this line have considered a small number (at most 5 or so) of layers in SFNN. We aim for the same goal, but our direction is orthogonal to them.

Instead of training SFNN directly, we study whether pre-trained parameters of DNN (or easier models) can be transferred to it, possibly with further fine-tuning of light cost. This approach can be attractive since one can utilize recent advances in DNN on its design and training. For example, one can design the network structure of SFNN following known specialized ones of DNN and use their pre-trained parameters. To this end, we first try transferring pre-trained parameters of DNN using sigmoid activation functions to those of the corresponding SFNN directly. In our experiments, the heuristic reasonably works well. For multi-modal learning, SFNN under such a simple transformation outperforms DNN. Even for the MNIST classification, the former performs similarly as the latter (see Section 2 for more details). However, it is questionable whether a similar strategy works in general, particularly for other unbounded activation functions like ReLU (Nair & Hinton, 2010) since SFNN has binary, i.e., bounded, random latent units. Moreover, it lost the regularization benefit of SFNN: it is rather believed that transferring parameters of stochastic models to DNN helps its regularization, but the opposite direction is unlikely possible.

To address the issues, we propose a special form of stochastic neural networks, named Simplified-SFNN, which intermediates between SFNN and DNN, having the following properties. First, Simplified-SFNN can be built upon any baseline DNN, possibly having unbounded activation functions. The most significant part of our approach lies in providing rigorous *network knowledge transferring* (Chen et al., 2015) between Simplified-SFNN and DNN. In particular, we prove that parameters of DNN can be transformed to those of the corresponding Simplified-SFNN while preserving the performance, i.e., both represent the same mapping and features. Second, Simplified-SFNN approximates certain SFNN, better than DNN, by simplifying its upper latent units above stochastic ones using two different non-linear activation functions. Simplified-SFNN is much easier to train than SFNN while utilizing its stochastic nature for regularization.

The above connection DNN $\rightarrow$ Simplified-SFNN $\rightarrow$ SFNN naturally suggests the following training procedure for both SFNN and Simplified-SFNN: train a baseline DNN first and then fine-tune its corresponding Simplified-SFNN initialized by the transformed DNN parameters. The pre-training stage accelerates the training task since DNN is faster to train than Simplified-SFNN. In addition, one can also utilize known DNN training techniques such as dropout and batch normalization for fine-tuning Simplified-SFNN. In our experiments, we train SFNN and Simplified-SFNN under the proposed strategy. They consistently outperform the corresponding DNN for both multi-modal and classification tasks, where the former and the latter are for measuring the model expressive power and the regularization effect, respectively. To the best of our knowledge, we are the first to confirm that SFNN indeed regularizes better than DNN. We also construct the stochastic models following the same network structure of popular DNNs including Lenet-5 (LeCun et al., 1998), NIN (Lin et al., 2014) and WRN (Zagoruyko & Komodakis, 2016). In particular, WRN (wide residual network) of 28 layers and 36 million parameters has shown the state-of-art performances on CIFAR-10 and CIFAR-100 classification datasets, and our stochastic models built upon WRN outperform the deterministic WRN on the datasets.

**Organization.** In Section 2, we focus on DNNs having sigmoid and ReLU activation functions and study simple transformations of their parameters to those of SFNN. In Section 3, we consider DNNs having general activation functions and describe more advanced transformations via introducing a new model, named Simplified-SFNN.

## 2 SIMPLE TRANSFORMATION FROM DNN TO SFNN

### 2.1 PRELIMINARIES FOR SFNN

Stochastic feedforward neural network (SFNN) is a hybrid model, which has both stochastic binary and deterministic hidden units. We first introduce SFNN with one stochastic hidden layer (and without deterministic hidden layers) for simplicity. Throughout this paper, we commonly denote the bias for unit $i$ and the weight matrix of the $\ell$-th hidden layer by $b_i^\ell$ and $\mathbf{W}^\ell$, respectively. Then, the stochastic hidden layer in SFNN is defined as a binary random vector with $N^1$ units, i.e., $\mathbf{h}^1 \in$

$\{0, 1\}^{N^1}$, drawn under the following distribution:

$$P\left(\mathbf{h}^1 \mid \mathbf{x}\right) = \prod_{i=1}^{N^1} P\left(h_i^1 \mid \mathbf{x}\right), \qquad \text{where} \quad P\left(h_i^1 = 1 \mid \mathbf{x}\right) = \sigma\left(\mathbf{W}_i^1 \mathbf{x} + b_i^1\right). \qquad (1)$$

In the above, $\mathbf{x}$ is the input vector and $\sigma\left(x\right) = 1 / \left(1 + e^{-x}\right)$ is the sigmoid function. Our conditional distribution of the output $y$ is defined as follows:

$$P\left(y \mid \mathbf{x}\right) = \mathbb{E}_{P(\mathbf{h}^1 | \mathbf{x})}\left[P\left(y \mid \mathbf{h}^1\right)\right] = \mathbb{E}_{P(\mathbf{h}^1 | \mathbf{x})}\left[\mathcal{N}\left(y \mid \mathbf{W}^2\mathbf{h}^1 + b^2,\ \sigma_y^2\right)\right],$$

where $\mathcal{N}(\cdot)$ denotes the normal distribution with mean $\mathbf{W}^2\mathbf{h}^1 + b^2$ and (fixed) variance $\sigma_y^2$. Therefore, $P\left(y \mid \mathbf{x}\right)$ can express a very complex, multi-modal distribution since it is a mixture of exponentially many normal distributions. The multi-layer extension is straightforward via a combination of stochastic and deterministic hidden layers, e.g., see Tang & Salakhutdinov (2013), Raiko et al. (2014). Furthermore, one can use any other output distributions as like DNN, e.g., softmax for classification tasks.

There are two computational issues for training SFNN: computing expectations with respect to stochastic units in forward pass and computing gradients in backward pass. One can notice that both are computationally intractable since they require summations over exponentially many configurations of all stochastic units. First, in order to handle the issue in forward pass, one can use the following Monte Carlo approximation for estimating the expectation: $P\left(y \mid \mathbf{x}\right) \simeq \frac{1}{M} \sum_{m=1}^{M} P(y \mid \mathbf{h}^{(m)})$,

where $\mathbf{h}^{(m)} \sim P\left(\mathbf{h}^1 \mid \mathbf{x}\right)$ and $M$ is the number of samples. This random estimator is unbiased and has relatively low variance (Tang & Salakhutdinov, 2013) since its accuracy does not depend on the dimensionality of $\mathbf{h}^1$ and one can draw samples from the exact distribution. Next, in order to handle the issue in backward pass, Neal (1990) proposed a Gibbs sampling, but it is known that it often mixes poorly. Saul et al. (1996) proposed a variational learning based on the mean-field approximation, but it has additional parameters making the variational lower bound looser. More recently, several other techniques have been proposed including unbiased estimators of the variational bound using importance sampling (Tang & Salakhutdinov, 2013; Raiko et al., 2014) and biased/unbiased estimators of the gradient for approximating backpropagation (Bengio et al., 2013; Raiko et al., 2014; Gu et al., 2015).

## 2.2 Simple transformation from sigmoid-DNN and ReLU-DNN to SFNN

Despite the recent advances, training SFNN is still very slow compared to DNN due to the sampling procedures: in particular, it is notoriously hard to train SFNN when the network structure is deeper and wider. In order to handle these issues, we consider the following approximation:

$$
\begin{aligned}
P\left(y \mid \mathbf{x}\right) &= \mathbb{E}_{P(\mathbf{h}^1 | \mathbf{x})}\left[\mathcal{N}\left(y \mid \mathbf{W}^2\mathbf{h}^1 + b^2,\ \sigma_y^2\right)\right] \\
&\simeq \mathcal{N}\left(y \mid \mathbb{E}_{P(\mathbf{h}^1 | \mathbf{x})}\left[\mathbf{W}^2\mathbf{h}^1\right] + b^2,\ \sigma_y^2\right) = \mathcal{N}\left(y \mid \mathbf{W}^2\sigma\left(\mathbf{W}^1\mathbf{x} + \mathbf{b}^1\right) + b^2,\ \sigma_y^2\right). \quad (2)
\end{aligned}
$$

Note that the above approximation corresponds to replacing stochastic units by deterministic ones such that their hidden activation values are same as marginal distributions of stochastic units, i.e., SFNN can be approximated by DNN using sigmoid activation functions, say sigmoid-DNN. When there exist more latent layers above the stochastic one, one has to apply similar approximations to all of them, i.e., exchanging the orders of expectations and non-linear functions, for making DNN and SFNN are equivalent. Therefore, instead of training SFNN directly, one can try transferring pre-trained parameters of sigmoid-DNN to those of the corresponding SFNN directly: train sigmoid-DNN instead of SFNN, and replace deterministic units by stochastic ones for the inference purpose. Although such a strategy looks somewhat 'rude', it was often observed in the literature that it reasonably works well for SFNN (Raiko et al., 2014) and we also evaluate it as reported in Table 1. We also note that similar approximations appear in the context of dropout: it trains a stochastic model averaging exponentially many DNNs sharing parameters, but also approximates a single DNN well.

Now we investigate a similar transformation in the case when DNN uses the unbounded ReLU activation function, say ReLU-DNN. Many recent deep networks are of ReLU-DNN type due to the gradient vanishing problem, and their pre-trained parameters are often available. Although it is straightforward to build SFNN from sigmoid-DNN, it is less clear in this case since ReLU is

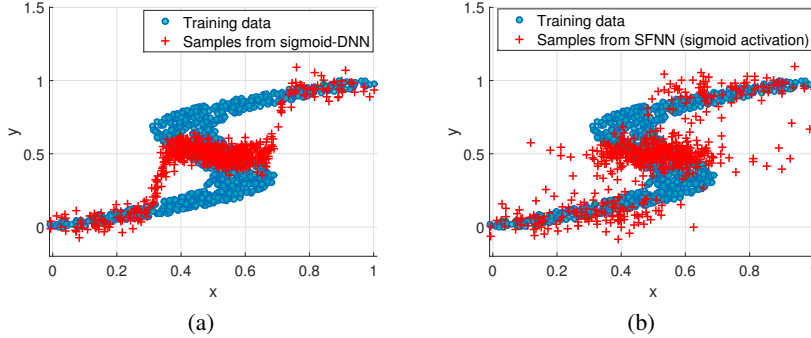

Figure 1: The generated samples from (a) sigmoid-DNN and (b) SFNN which uses same parameters trained by sigmoid-DNN. One can note that SFNN can model the multiple modes in outuput space $y$ around $x = 0.4$.

| Inference Model | Network Structure | MNIST Classification | | | Multi-modal Learning |
|---|---|---|---|---|---|
| | | Training NLL | Training Error (%) | Test Error (%) | Test NLL |
| sigmoid-DNN | 2 hidden layers | 0 | 0 | 1.54 | 5.290 |
| SFNN | 2 hidden layers | 0 | 0 | 1.56 | 1.564 |
| sigmoid-DNN | 3 hidden layers | 0.002 | 0.03 | 1.84 | 4.880 |
| SFNN | 3 hidden layers | 0.022 | 0.04 | 1.81 | 0.575 |
| sigmoid-DNN | 4 hidden layers | 0 | 0.01 | 1.74 | 4.850 |
| SFNN | 4 hidden layers | 0.003 | 0.03 | 1.73 | 0.392 |
| ReLU-DNN | 2 hidden layers | 0.005 | 0.04 | 1.49 | 7.492 |
| SFNN | 2 hidden layers | 0.819 | 4.50 | 5.73 | 2.678 |
| ReLU-DNN | 3 hidden layers | 0 | 0 | 1.43 | 7.526 |
| SFNN | 3 hidden layers | 1.174 | 16.14 | 17.83 | 4.468 |
| ReLU-DNN | 4 hidden layers | 0 | 0 | 1.49 | 7.572 |
| SFNN | 4 hidden layers | 1.213 | 13.13 | 14.64 | 1.470 |

Table 1: The performance of simple parameter transformations from DNN to SFNN on the MNIST and synthetic datasets, where each layer of neural networks contains 800 and 50 hidden units for two datasets, respectively. For all experiments, the only first hidden layer of DNN is replaced by stochastic one. We report negative log-likelihood (NLL) and classification error rates.

unbounded. To handle this issue, we redefine the stochastic latent units of SFNN:

$$P\left(\mathbf{h}^1 \mid \mathbf{x}\right) = \prod_{i=1}^{N^1} P\left(h_i^1 \mid \mathbf{x}\right), \qquad \text{where} \quad P\left(h_i^1 = 1 \mid \mathbf{x}\right) = \min\left\{\alpha f\left(\mathbf{W}_i^1 \mathbf{x} + b_i^1\right), 1\right\}. \quad (3)$$

In the above, $f(x) = \max\{x, 0\}$ is the ReLU activation function and $\alpha$ is some hyper-parameter. A simple transformation can be defined similarly as the case of sigmoid-DNN via replacing deterministic units by stochastic ones. However, to preserve the parameter information of ReLU-DNN, one has to choose $\alpha$ such that $\alpha f\left(\mathbf{W}_i^1 \mathbf{x} + b_i^1\right) \leq 1$ and rescale upper parameters $\mathbf{W}^2$ as follows:

$$\alpha^{-1} \leftarrow \max_{i,\mathbf{x}} \left| f\left(\widehat{\mathbf{W}}_i^1 \mathbf{x} + \widehat{b}_i^1\right)\right|, \quad \left(\mathbf{W}^1, \mathbf{b}^1\right) \leftarrow \left(\widehat{\mathbf{W}}^1, \widehat{\mathbf{b}}^1\right), \quad \left(\mathbf{W}^2, \mathbf{b}^2\right) \leftarrow \left(\widehat{\mathbf{W}}^2/\alpha, \widehat{\mathbf{b}}^2\right). \quad (4)$$

Then, applying similar approximations as in (2), i.e., exchanging the orders of expectations and non-linear functions, one can observe that ReLU-DNN and SFNN are equivalent.

We evaluate the performance of the simple transformations from DNN to SFNN on the MNIST dataset (LeCun et al., 1998) and the synthetic dataset (Bishop, 1994), where the former and the latter are popular datasets used for a classification task and a multi-modal (i.e., one-to-many mappings) prediction learning, respectively. In all experiments reported in this paper, we commonly use the softmax and Gaussian with standard deviation of $\sigma_y = 0.05$ are used for the output probability on classification and regression tasks, respectively. The only first hidden layer of DNN is replaced by stochastic one, and we use 500 samples for estimating the expectations in the SFNN inference. As reported in Table 1, we observe that the simple transformation often works well for both tasks: the SFNN and sigmoid-DNN inferences (using same parameters trained by sigmoid-DNN) perform similarly for the classification task and the former significantly outperforms for the latter for the

multi-modal task (also see Figure 1). It might suggest some possibilities that the expensive SFNN training might not be not necessary, depending on the targeted learning quality. However, in case of ReLU, SFNN performs much worse than ReLU-DNN for the MNIST classification task under the parameter transformation.

## 3 TRANSFORMATION FROM DNN TO SFNN VIA SIMPLIFIED-SFNN

In this section, we propose an advanced method to utilize the pre-trained parameters of DNN for training SFNN. As shown in the previous section, simple parameter transformations from DNN to SFNN are not clear to work in general, in particular for activation functions other than sigmoid. Moreover, training DNN does not utilize the stochastic regularizing effect, which is an important benefit of SFNN. To address the issues, we design an intermediate model, called Simplified-SFNN. The proposed model is a special form of stochastic neural networks, which approximates certain SFNN by simplifying its upper latent units above stochastic ones. Then, we establish more rigorous connections between three models: DNN → Simplified-SFNN → SFNN, which leads to an efficient training procedure of the stochastic models utilizing pre-trained parameters of DNN. In our experiments, we evaluate the strategy for various tasks and popular DNN architectures.

### 3.1 SIMPLIFIED-SFNN OF TWO HIDDEN LAYERS AND NON-NEGATIVE ACTIVATION FUNCTIONS

For clarity of presentation, we first introduce Simplified-SFNN with two hidden layers and non-negative activation functions, where its extensions to multiple layers and general activation functions are presented in Appendix B. We also remark that we primarily describe fully-connected Simplified-SFNNs, but their convolutional versions can also be naturally defined. In Simplified-SFNN of two hidden layers, we assume that the first and second hidden layers consist of stochastic binary hidden units and deterministic ones, respectively. As like (3), the first layer is defined as a binary random vector with $N^1$ units, i.e., $\mathbf{h}^1 \in \{0, 1\}^{N^1}$, drawn under the following distribution:

$$P\left(\mathbf{h}^1 \mid \mathbf{x}\right) = \prod_{i=1}^{N^1} P\left(h_i^1 \mid \mathbf{x}\right), \qquad \text{where} \quad P\left(h_i^1 = 1 \mid \mathbf{x}\right) = \min\left\{\alpha_1 f\left(\mathbf{W}_i^1 \mathbf{x} + b_i^1\right), 1\right\}. \quad (5)$$

where $\mathbf{x}$ is the input vector, $\alpha_1 > 0$ is a hyper-parameter for the first layer, and $f : \mathbb{R} \to \mathbb{R}_+$ is some non-negative non-linear activation function with $|f'(x)| \leq 1$ for all $x \in \mathbb{R}$, e.g., ReLU and sigmoid activation functions. Now the second layer is defined as the following deterministic vector with $N^2$ units, i.e., $\mathbf{h}^2(\mathbf{x}) \in \mathbb{R}^{N^2}$:

$$\mathbf{h}^2\left(\mathbf{x}\right) = \left[f\left(\alpha_2\left(\mathbb{E}_{P(\mathbf{h}^1 \mid \mathbf{x})}\left[s\left(\mathbf{W}_j^2 \mathbf{h}^1 + b_j^2\right)\right] - s\left(0\right)\right)\right) : \forall j \in N^2\right], \quad (6)$$

where $\alpha_2 > 0$ is a hyper-parameter for the second layer and $s : \mathbb{R} \to \mathbb{R}$ is a differentiable function with $|s''(x)| \leq 1$ for all $x \in \mathbb{R}$, e.g., sigmoid and tanh functions. In our experiments, we use the sigmoid function for $s(x)$. Here, one can note that the proposed model also has the same computational issues with SFNN in forward and backward passes due to the complex expectation. One can train Simplified-SFNN similarly as SFNN: we use Monte Carlo approximation for estimating the expectation and the (biased) estimator of the gradient for approximating backpropagation inspired by Raiko et al. (2014) (more detailed explanation is presented in Appendix A).

We are interested in transferring parameters of DNN to Simplified-SFNN to utilize the training benefits of DNN since the former is much faster to train than the latter. To this end, we consider the following DNN of which $\ell$-th hidden layer is deterministic and defined as follows:

$$\widehat{\mathbf{h}}^\ell\left(\mathbf{x}\right) = \left[\widehat{h}_i^\ell\left(\mathbf{x}\right) = f\left(\widehat{\mathbf{W}}_i^\ell \widehat{\mathbf{h}}^{\ell-1}\left(\mathbf{x}\right) + \widehat{b}_i^\ell\right) : i \in N^\ell\right], \quad (7)$$

where $\widehat{\mathbf{h}}^0(\mathbf{x}) = \mathbf{x}$. As stated in the following theorem, we establish a rigorous way how to initialize parameters of Simplified-SFNN in order to transfer the knowledge stored in DNN.

**Theorem 1** *Assume that both DNN and Simplified-SFNN with two hidden layers have same network structure with non-negative activation function $f$. Given parameters $\{\widehat{\mathbf{W}}^\ell, \widehat{\mathbf{b}}^\ell : \ell = 1, 2\}$ of DNN and input dataset $D$, choose those of Simplified-SFNN as follows:*

$$\left(\alpha_1, \mathbf{W}^1, \mathbf{b}^1\right) \leftarrow \left(\frac{1}{\gamma_1}, \widehat{\mathbf{W}}^1, \widehat{\mathbf{b}}^1\right), \quad \left(\alpha_2, \mathbf{W}^2, \mathbf{b}^2\right) \leftarrow \left(\frac{\gamma_2 \gamma_1}{s'(0)}, \frac{1}{\gamma_2}\widehat{\mathbf{W}}^2, \frac{1}{\gamma_1 \gamma_2}\widehat{\mathbf{b}}^2\right), \quad (8)$$

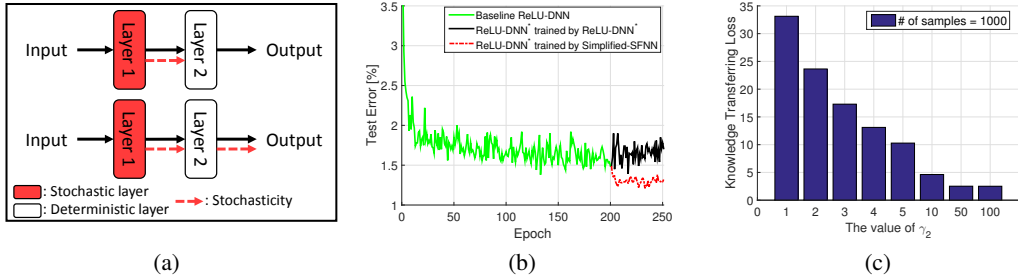

(a) (b) (c)

Figure 2: (a) Simplified-SFNN (top) and SFNN (bottom). (b) For first 200 epochs, we train a baseline ReLU-DNN. Then, we train simplified-SFNN initialized by the DNN parameters under transformation (8) with $\gamma_2 = 50$. We observe that training ReLU-DNN* directly does not reduce the test error even when network knowledge transferring still holds between the baseline ReLU-DNN and the corresponding ReLU-DNN*. (c) As the value of $\gamma_2$ increases, knowledge transferring loss measured as $\frac{1}{|D|} \frac{1}{N^\ell} \sum_{\mathbf{x}} \sum_i \left| h_i^\ell (\mathbf{x}) - \widehat{h}_i^\ell (\mathbf{x}) \right|$ is decreasing.

*where* $\gamma_1 = \max_{i, \mathbf{x} \in D} \left| f \left( \widehat{\mathbf{W}}_i^1 \mathbf{x} + \widehat{b}_i^1 \right) \right|$ *and* $\gamma_2 > 0$ *is any positive constant. Then, it follows that*

$$\left| h_j^2 (\mathbf{x}) - \widehat{h}_j^2 (\mathbf{x}) \right| \leq \frac{\gamma_1 \left( \sum_i \left| \widehat{W}_{ij}^2 \right| + \widehat{b}_j^2 \gamma_1^{-1} \right)^2}{2 s' (0) \gamma_2}, \quad \forall j, \mathbf{x} \in D.$$

The proof of the above theorem is presented in Appendix D.1. Our proof is built upon the first-order Taylor expansion of non-linear function $s(x)$. Theorem 1 implies that one can make Simplified-SFNN represent the function values of DNN with bounded errors using a linear transformation. Furthermore, the errors can be made arbitrarily small by choosing large $\gamma_2$, i.e., $\lim_{\gamma_2 \to \infty} \left| h_j^2 (\mathbf{x}) - \widehat{h}_j^2 (\mathbf{x}) \right| = 0, \ \forall j, \mathbf{x} \in D$. Figure 2(c) shows that knowledge transferring loss decreases as $\gamma_2$ increases on MNIST classification. Based on this, we choose $\gamma_2 = 50$ commonly for all experiments.

## 3.2 WHY SIMPLIFIED-SFNN ?

Given a Simplified-SFNN model, the corresponding SFNN can be naturally defined by taking out the expectation in (6). As illustrated in Figure 2(a), the main difference between SFNN and Simplified-SFNN is that the randomness of the stochastic layer propagates only to its upper layer in the latter, i.e., the randomness of $\mathbf{h}^1$ is averaged out at its upper units $\mathbf{h}^2$ and does not propagate to $\mathbf{h}^3$ or output $y$. Hence, Simplified-SFNN is no longer a Bayesian network. This makes training Simplified-SFNN much easier than SFNN since random samples are not required at some layers[1] and consequently the quality of gradient estimations can also be improved, in particular for unbounded activation functions. Furthermore, one can use the same approximation procedure (2) to see that Simplified-SFNN approximates SFNN. However, since Simplified-SFNN still maintains binary random units, it uses approximation steps later, in comparison with DNN. In summary, Simplified-SFNN is an intermediate model between DNN and SFNN, i.e., DNN $\to$ Simplified-SFNN $\to$ SFNN.

The above connection naturally suggests the following training procedure for both SFNN and Simplified-SFNN: train a baseline DNN first and then fine-tune its corresponding Simplified-SFNN initialized by the transformed DNN parameters. Finally, the fine-tuned parameters can be used for SFNN as well. We evaluate the strategy for the MNIST classification, which is reported in Table 2 (see Appendix C for more detailed experiment setups). We found that SFNN under the two-stage training always performs better than SFNN under a simple transformation (4) from ReLU-DNN.

---

[1] For example, if one replaces the first feature maps in the fifth residual unit of Pre-ResNet having 164 layers (He et al., 2016) by stochastic ones, then the corresponding DNN, Simplified-SFNN and SFNN took 1 mins 35 secs, 2 mins 52 secs and 16 mins 26 secs per each training epoch, respectively, on our machine with one Intel CPU (Core i7-5820K 6-Core@3.3GHz) and one NVIDIA GPU (GTX Titan X, 3072 CUDA cores). Here, we trained both stochastic models using the biased estimator (Raiko et al., 2014) with 10 random samples on CIFAR-10 dataset.

| Inference Model | Training Model | Network Structure | without BN & DO | with BN | with DO |
|---|---|---|---|---|---|
| sigmoid-DNN | sigmoid-DNN | 2 hidden layers | 1.54 | 1.57 | 1.25 |
| SFNN | sigmoid-DNN | 2 hidden layers | 1.56 | 2.23 | 1.27 |
| Simplified-SFNN | fine-tuned by Simplified-SFNN | 2 hidden layers | 1.51 | 1.5 | 1.11 |
| sigmoid-DNN* | fine-tuned by Simplified-SFNN | 2 hidden layers | 1.48 (0.06) | 1.48 (0.09) | 1.14 (0.11) |
| SFNN | fine-tuned by Simplified-SFNN | 2 hidden layers | 1.51 | 1.57 | 1.11 |
| ReLU-DNN | ReLU-DNN | 2 hidden layers | 1.49 | 1.25 | 1.12 |
| SFNN | ReLU-DNN | 2 hidden layers | 5.73 | 3.47 | 1.74 |
| Simplified-SFNN | fine-tuned by Simplified-SFNN | 2 hidden layers | 1.41 | 1.17 | 1.06 |
| ReLU-DNN* | fine-tuned by Simplified-SFNN | 2 hidden layers | 1.32 (0.17) | 1.16 (0.09) | 1.05 (0.07) |
| SFNN | fine-tuned by Simplified-SFNN | 2 hidden layers | 2.63 | 1.34 | 1.51 |
| ReLU-DNN | ReLU-DNN | 3 hidden layers | 1.43 | 1.34 | 1.24 |
| SFNN | ReLU-DNN | 3 hidden layers | 17.83 | 4.15 | 1.49 |
| Simplified-SFNN | fine-tuned by Simplified-SFNN | 3 hidden layers | 1.28 | 1.25 | 1.04 |
| ReLU-DNN* | fine-tuned by Simplified-SFNN | 3 hidden layers | 1.27 (0.16) | 1.24 (0.1) | **1.03 (0.21)** |
| SFNN | fine-tuned by Simplified-SFNN | 3 hidden layers | 1.56 | 1.82 | 1.16 |
| ReLU-DNN | ReLU-DNN | 4 hidden layers | 1.49 | 1.34 | 1.30 |
| SFNN | ReLU-DNN | 4 hidden layers | 14.64 | 3.85 | 2.17 |
| Simplified-SFNN | fine-tuned by Simplified-SFNN | 4 hidden layers | 1.32 | 1.32 | 1.25 |
| ReLU-DNN* | fine-tuned by Simplified-SFNN | 4 hidden layers | 1.29 (0.2) | 1.29 (0.05) | 1.25 (0.05) |
| SFNN | fine-tuned by Simplified-SFNN | 4 hidden layers | 3.44 | 1.89 | 1.56 |

Table 2: Classification test error rates [%] on MNIST, where each layer of neural networks contains 800 hidden units. All Simplified-SFNNs are constructed by replacing the first hidden layer of a baseline DNN with stochastic hidden layer. We also consider training DNN and fine-tuning Simplified-SFNN using batch normalization (BN) and dropout (DO). The performance improvements beyond baseline DNN (due to fine-tuning DNN parameters under Simplified-SFNN) are calculated in the bracket.

More interestingly, Simplified-SFNN consistently outperforms its baseline DNN due to the stochastic regularizing effect, even when we train both models using dropout (Hinton et al., 2012b) and batch normalization (Ioffe & Szegedy, 2015). In order to confirm the regularization effects, one can again approximate a trained Simplified-SFNN by a new deterministic DNN which we call DNN* and is different from its baseline DNN under the following approximation at upper latent units above binary random units:

$$\mathbb{E}_{P(\mathbf{h}^\ell|\mathbf{x})}\left[s\left(\mathbf{W}_j^{\ell+1}\mathbf{h}^\ell\right)\right] \simeq s\left(\mathbb{E}_{P(\mathbf{h}^\ell|\mathbf{x})}\left[\mathbf{W}_j^{\ell+1}\mathbf{h}^\ell\right]\right) = s\left(\sum_i W_{ij}^{\ell+1}P\left(h_i^\ell = 1 \mid \mathbf{x}\right)\right). \quad (9)$$

We found that DNN* using fined-tuned parameters of Simplified-SFNN also outperforms the baseline DNN as shown in Table 2 and Figure 2(b).

### 3.3 EXPERIMENTAL RESULTS ON MULTI-MODAL LEARNING AND CONVOLUTIONAL NETWORKS

We present several experimental results for both multi-modal and classification tasks on MNIST (LeCun et al., 1998), Toronto Face Database (TFD) (Susskind et al., 2010), CIFAR-10, CIFAR-100 (Krizhevsky & Hinton, 2009) and SVHN (Netzer et al., 2011). Here, we present some key results due to the space constraints and more detailed explanations for our experiment setups are presented in Appendix C.

We first verify that it is possible to learn one-to-many mapping via Simplified-SFNN on the TFD and MNIST datasets, where the former and the latter are used to predict multiple facial expressions from the mean of face images per individual and the lower half of the MNIST digit given the upper half, respectively. We remark that both tasks are commonly performed in recent other works to test the multi-modal learning using SFNN (Raiko et al., 2014; Gu et al., 2015). In all experiments, we first train a baseline DNN, and the trained parameters of DNN are used for further fine-tuning those of Simplified-SFNN. As shown in Table 3 and Figure 3, stochastic models outperform their baseline DNN, and generate different digits for the case of ambiguous inputs (while DNN does not). We also evaluate the regularization effect of Simplified-SFNN for the classification tasks on CIFAR-10, CIFAR-100 and SVHN. Table 4 reports the classification error rates using convolutional neural networks such as Lenet-5 (LeCun et al., 1998), NIN (Lin et al., 2014) and WRN (Zagoruyko & Komodakis, 2016). Due to the regularization effects, Simplified-SFNNs consistently outperform

| Inference Model | Training Model | MNIST-half | | TFD | |
|---|---|---|---|---|---|
| | | 2 hidden layers | 3 hidden layers | 2 hidden layers | 3 hidden layers |
| sigmoid-DNN | sigmoid-DNN | 1.409 | 1.720 | -0.064 | 0.005 |
| SFNN | sigmoid-DNN | 0.644 | 1.076 | -0.461 | -0.401 |
| Simplified-SFNN | fine-tuned by Simplified-SFNN | 1.474 | 1.757 | -0.071 | -0.028 |
| SFNN | fine-tuned by Simplified-SFNN | 0.619 | 0.991 | **-0.509** | -0.423 |
| ReLU-DNN | ReLU-DNN | 1.747 | 1.741 | 1.271 | 1.232 |
| SFNN | ReLU-DNN | -1.019 | -1.021 | 0.823 | 1.121 |
| Simplified-SFNN | fine-tuned by Simplified-SFNN | 2.122 | 2.226 | 0.175 | 0.343 |
| SFNN | fine-tuned by Simplified-SFNN | **-1.290** | -1.061 | -0.380 | -0.193 |

Table 3: Test negative log-likelihood (NLL) on MNIST-half and TFD datasets, where each layer of neural networks contains 200 hidden units. All Simplified-SFNNs are constructed by replacing the first hidden layer of a baseline DNN with stochastic hidden layer.

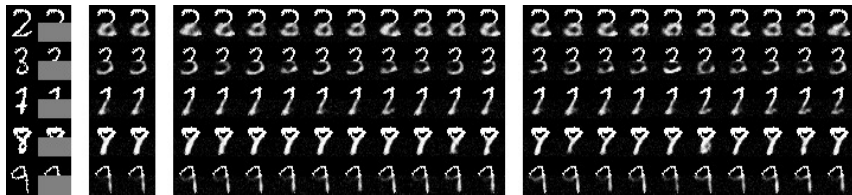

Figure 3: Generated samples for predicting the lower half of the MNIST digit given the upper half. The original digits and the corresponding inputs (first). The generated samples from sigmoid-DNN (second), SFNN under the simple transformation (third), and SFNN fine-tuned by Simplified-SFNN (forth). We observed that SFNN fine-tuned by Simplified-SFNN can generate more various samples from same inputs, e.g., 3 and 8, better than SFNN under the simple transformation.

| Inference model | Training Model | CIFAR-10 | CIFAR-100 | SVHN |
|---|---|---|---|---|
| Lenet-5 | Lenet-5 | 37.67 | 77.26 | 11.18 |
| Lenet-5* | Simplified-SFNN | 33.58 | 73.02 | 9.88 |
| NIN | NIN | 9.51 | 32.66 | 3.21 |
| NIN* | Simplified-SFNN | 9.33 | 30.81 | 3.01 |
| WRN | WRN | 4.22 (4.39)† | 20.30 (20.04)† | 3.25† |
| WRN* | Simplified-SFNN (one stochastic layer) | 4.21† | 19.98† | 3.09† |
| WRN* | Simplified-SFNN (two stochastic layers) | **4.14**† | **19.72**† | 3.06† |

Table 4: Test error rates [%] on CIFAR-10, CIFAR-100 and SVHN. The error rates for WRN are from our experiments, where original ones reported in (Zagoruyko & Komodakis, 2016) are in the brackets. Results with † are obtained using the horizontal flipping and random cropping augmentation.

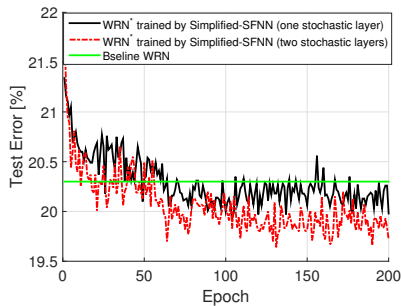

Figure 4: Test errors of WRN* per each training epoch on CIFAR-100.

their baseline DNNs. For example, WRN* outperforms WRN by 0.08% on CIFAR-10 and 0.58% on CIFAR-100. We expect that if one introduces more stochastic layers, the error would be decreased more (see Figure 4), but it increases the fine-tuning time-complexity of Simplified-SFNN.

## 4 CONCLUSION

In order to develop an efficient training method for large-scale SFNN, this paper proposes a new intermediate stochastic model, called Simplified-SFNN. We establish the connection between three models, i.e., DNN → Simplified-SFNN → SFNN, which naturally leads to an efficient training procedure of the stochastic models utilizing pre-trained parameters of DNN. This connection naturally leads an efficient training procedure of the stochastic models utilizing pre-trained parameters and architectures of DNN. We believe that our work brings a new important direction for training stochastic neural networks, which should be of broader interest in many related applications.

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

# A  TRAINING SIMPLIFIED-SFNN

The parameters of Simplified-SFNN can be learned using a variant of the backpropagation algorithm (Rumelhart et al., 1988) in a similar manner to DNN. However, in contrast to DNN, there are two computational issues for simplified-SFNN: computing expectations with respect to stochastic units in forward pass and computing gradients in back pass. One can notice that both are intractable since they require summations over all possible configurations of all stochastic units. First, in order to handle the issue in forward pass, we use the following Monte Carlo approximation for estimating the expectation:

$$\mathbb{E}_{P(\mathbf{h}^1|\mathbf{x})}\left[s\left(\mathbf{W}_j^2\mathbf{h}^1 + b_j^2\right)\right] \simeq \frac{1}{M}\sum_{m=1}^{M} s\left(\mathbf{W}_j^2\mathbf{h}^{(m)} + b_j^2\right), \qquad \mathbf{h}^{(m)} \sim P\left(\mathbf{h}^1 \mid \mathbf{x}\right),$$

where $M$ is the number of samples. This random estimator is unbiased and has relatively low variance (Tang & Salakhutdinov, 2013) since its accuracy does not depend on the dimensionality of $\mathbf{h}^1$ and one can draw samples from the exact distribution. Next, in order to handle the issue in back pass, we use the following approximation inspired by (Raiko et al., 2014):

$$\frac{\partial}{\partial \mathbf{W}_j^2}\mathbb{E}_{P(\mathbf{h}^1|\mathbf{x})}\left[s\left(\mathbf{W}_j^2\mathbf{h}^1 + b_j^2\right)\right] \simeq \frac{1}{M}\sum_{m}\frac{\partial}{\partial \mathbf{W}_j^2}s\left(\mathbf{W}_j^2\mathbf{h}^{(m)} + b_j^2\right),$$

$$\frac{\partial}{\partial \mathbf{W}_i^1}\mathbb{E}_{P(\mathbf{h}^1|\mathbf{x})}\left[s\left(\mathbf{W}_j^2\mathbf{h}^1 + b_j^2\right)\right] \simeq \frac{W_{ij}^2}{M}\sum_{m}s'\left(\mathbf{W}_j^2\mathbf{h}^{(m)} + b_j^2\right)\frac{\partial}{\partial \mathbf{W}_i^1}P\left(h_i^1 = 1 \mid \mathbf{x}\right),$$

where $\mathbf{h}^{(m)} \sim P\left(\mathbf{h}^1 \mid \mathbf{x}\right)$ and $M$ is the number of samples. In our experiments, we commonly choose $M = 20$.

# B  EXTENSIONS OF SIMPLIFIED-SFNN

In this section, we describe how the network knowledge transferring between Simplified-SFNN and DNN, i.e., Theorem 1, generalizes to multiple layers and general activation functions.

## B.1  EXTENSION TO MULTIPLE LAYERS

A deeper Simplified-SFNN with $L$ hidden layers can be defined similarly as the case of $L = 2$. We also establish network knowledge transferring between Simplified-SFNN and DNN with $L$ hidden layers as stated in the following theorem. Here, we assume that stochastic layers are not consecutive for simpler presentation, but the theorem is generalizable for consecutive stochastic layers.

**Theorem 2** *Assume that both DNN and Simplified-SFNN with $L$ hidden layers have same network structure with non-negative activation function $f$. Given parameters $\{\widehat{\mathbf{W}}^\ell, \widehat{\mathbf{b}}^\ell : \ell = 1, \ldots, L\}$ of DNN and input dataset $D$, choose the same ones for Simplified-SFNN initially and modify them for each $\ell$-th stochastic layer and its upper layer as follows:*

$$\alpha_\ell \leftarrow \frac{1}{\gamma_\ell}, \quad \left(\alpha_{\ell+1}, \mathbf{W}^{\ell+1}, \mathbf{b}^{\ell+1}\right) \leftarrow \left(\frac{\gamma_\ell \gamma_{\ell+1}}{s'(0)}, \frac{\widehat{\mathbf{W}}^{\ell+1}}{\gamma_{\ell+1}}, \frac{\widehat{\mathbf{b}}^{\ell+1}}{\gamma_\ell \gamma_{\ell+1}}\right),$$

*where $\gamma_\ell = \max\limits_{i, \mathbf{x} \in D} \left| f\left(\widehat{\mathbf{W}}_i^\ell \mathbf{h}^{\ell-1}(\mathbf{x}) + \widehat{b}_i^\ell\right) \right|$ and $\gamma_{\ell+1}$ is any positive constant. Then, it follows that*

$$\lim_{\substack{\gamma_{\ell+1} \to \infty \\ \forall \text{ stochastic hidden layer } \ell}} \left| h_j^L(\mathbf{x}) - \widehat{h}_j^L(\mathbf{x}) \right| = 0, \quad \forall j, \mathbf{x} \in D.$$

The above theorem again implies that it is possible to transfer knowledge from DNN to Simplified-SFNN by choosing large $\gamma_{l+1}$. The proof of Theorem 2 is similar to that of Theorem 1 and given in Appendix D.2.

### B.2 EXTENSION TO GENERAL ACTIVATION FUNCTIONS

In this section, we describe an extended version of Simplified-SFNN which can utilize any activation function. To this end, we modify the definitions of stochastic layers and their upper layers by introducing certain additional terms. If the $\ell$-th hidden layer is stochastic, then we slightly modify the original definition (5) as follows:

$$P\left(\mathbf{h}^\ell \mid \mathbf{x}\right) = \prod_{i=1}^{N^\ell} P\left(h_i^\ell \mid \mathbf{x}\right) \quad \text{with} \quad P\left(h_i^\ell = 1 \mid \mathbf{x}\right) = \min\left\{\alpha_\ell f\left(\mathbf{W}_i^1 \mathbf{x} + b_i^1 + \frac{1}{2}\right), 1\right\},$$

where $f : \mathbb{R} \to \mathbb{R}$ is a non-linear (possibly, negative) activation function with $|f'(x)| \leq 1$ for all $x \in \mathbb{R}$. In addition, we re-define its upper layer as follows:

$$\mathbf{h}^{\ell+1}(\mathbf{x}) = \left[ f\left( \alpha_{\ell+1} \left( \mathbb{E}_{P(\mathbf{h}^\ell \mid \mathbf{x})} \left[ s\left( \mathbf{W}_j^{\ell+1} \mathbf{h}^\ell + b_j^{\ell+1} \right) \right] - s(0) - \frac{s'(0)}{2} \sum_i W_{ij}^{\ell+1} \right) \right) : \forall j \right],$$

where $\mathbf{h}^0(\mathbf{x}) = \mathbf{x}$ and $s : \mathbb{R} \to \mathbb{R}$ is a differentiable function with $|s''(x)| \leq 1$ for all $x \in \mathbb{R}$.

Under this general Simplified-SFNN model, we also show that transferring network knowledge from DNN to Simplified-SFNN is possible as stated in the following theorem. Here, we again assume that stochastic layers are not consecutive for simpler presentation.

**Theorem 3** *Assume that both DNN and Simplified-SFNN with $L$ hidden layers have same network structure with non-linear activation function $f$. Given parameters $\{\widehat{\mathbf{W}}^\ell, \widehat{\mathbf{b}}^\ell : \ell = 1, \ldots, L\}$ of DNN and input dataset $D$, choose the same ones for Simplified-SFNN initially and modify them for each $\ell$-th stochastic layer and its upper layer as follows:*

$$\alpha_\ell \leftarrow \frac{1}{2\gamma_\ell}, \quad \left(\alpha_{\ell+1}, \mathbf{W}^{\ell+1}, \mathbf{b}^{\ell+1}\right) \leftarrow \left(\frac{2\gamma_\ell \gamma_{\ell+1}}{s'(0)}, \frac{\widehat{\mathbf{W}}^{\ell+1}}{\gamma_{\ell+1}}, \frac{\widehat{\mathbf{b}}^{\ell+1}}{2\gamma_\ell \gamma_{\ell+1}}\right),$$

*where $\gamma_\ell = \max\limits_{i, \mathbf{x} \in D} \left| f\left(\widehat{\mathbf{W}}_i^\ell \mathbf{h}^{\ell-1}(\mathbf{x}) + \widehat{b}_i^\ell\right) \right|$, and $\gamma_{\ell+1}$ is any positive constant. Then, it follows that*

$$\lim_{\substack{\gamma_{\ell+1} \to \infty \\ \forall \text{ stochastic hidden layer } \ell}} \left| h_j^L(\mathbf{x}) - \widehat{h}_j^L(\mathbf{x}) \right| = 0, \quad \forall j, \mathbf{x} \in D.$$

We omit the proof of the above theorem since it is somewhat direct adaptation of that of Theorem 2.

## C EXPERIMENTAL SETUPS

In this section, we describe detailed explanation about all the experiments described in Section 3. In all experiments, the softmax and Gaussian with the standard deviation of 0.05 are used as the output probability for the classification task and the multi-modal prediction, respectively. The loss was minimized using ADAM learning rule (Kingma & Ba, 2014) with a mini-batch size of 128. We used an exponentially decaying learning rate.

## C.1 CLASSIFICATION ON MNIST

The MNIST dataset consists of $28 \times 28$ pixel greyscale images, each containing a digit 0 to 9 with 60,000 training and 10,000 test images. For this experiment, we do not use any data augmentation or pre-processing. Hyper-parameters are tuned on the validation set consisting of the last 10,000 training images. All Simplified-SFNNs are constructed by replacing the first hidden layer of a baseline DNN with stochastic hidden layer. As described in Section 3.2, we train Simplified-SFNNs under the two-stage procedure: first train a baseline DNN for first 200 epochs, and the trained parameters of DNN are used for initializing those of Simplified-SFNN. For 50 epochs, we train simplified-SFNN. We choose the hyper-parameter $\gamma_2 = 50$ in the parameter transformation. All Simplified-SFNNs are trained with $M = 20$ samples at each epoch, and in the test, we use 500 samples.

## C.2 MULTI-MODAL REGRESSION ON TFD AND MNIST

The Toronto Face Database (TFD) (Susskind et al., 2010) dataset consists of $48 \times 48$ pixel greyscale images, each containing a face image of 900 individuals with 7 different expressions. Similar to (Raiko et al., 2014), we use 124 individuals with at least 10 facial expressions as data. We randomly choose 100 individuals with 1403 images for training and the remaining 24 individuals with 326 images for the test. We take the mean of face images per individual as the input and set the output as the different expressions of the same individual. The MNIST dataset consists of $28 \times 28$ pixel greyscale images, each containing a digit 0 to 9 with 60,000 training and 10,000 test images. For this experiments, each pixel of every digit images is binarized using its grey-scale value. We take the upper half of the MNIST digit as the input and set the output as the lower half of it. All Simplified-SFNNs are constructed by replacing the first hidden layer of a baseline DNN with stochastic hidden layer. We train Simplified-SFNNs with $M = 20$ samples at each epoch, and in the test, we use 500 samples. We use 200 hidden units for each layer of neural networks in two experiments. Learning rate is chosen from $\{0.005, 0.002, 0.001, ..., 0.0001\}$, and the best result is reported for both tasks.

## C.3 CLASSIFICATION ON CIFAR-10, CIFAR-100 AND SVHN

The CIFAR-10 and CIFAR-100 datasets consist of 50,000 training and 10,000 test images. The SVHN dataset consists of 73,257 training and 26,032 test images.[2] We pre-process the data using global contrast normalization and ZCA whitening. For these datasets, we design a convolutional version of Simplified-SFNN. In a similar manner to the case of fully-connected networks, one can define a stochastic convolution layer, which considers the input feature map as a binary random matrix and generates the output feature map as defined in (6). All Simplified-SFNNs are constructed by replacing a hidden feature map of a baseline models, i.e., Lenet-5, NIN and WRN, with stochastic one as shown in Figure 5(d). We use WRN with 16 and 28 layers for SVHN and CIFAR datasets, respectively, since they showed state-of-the-art performance as reported by Zagoruyko & Komodakis (2016). In case of WRN, we introduce up to two stochastic convolution layers. For 100 epochs, we first train baseline models, i.e., Lenet-5, NIN and WRN, and trained parameters are used for initializing those of Simplified-SFNNs. All Simplified-SFNNs are trained with $M = 5$ samples and the test error is only measured by the approximation (9). The test errors of baseline models are measured after training them for 200 epochs similar to Zagoruyko & Komodakis (2016).

# D  PROOFS OF THEOREMS

## D.1  PROOF OF THEOREM 1

First consider the first hidden layer, i.e., stochastic layer. Let $\gamma_1 = \max\limits_{i, \mathbf{x} \in D} f\left(\widehat{\mathbf{W}}_i^1 \mathbf{x} + \widehat{b}_i^1\right)$ be the maximum value of hidden units in DNN. If we initialize the parameters $(\alpha_1, \mathbf{W}^1, \mathbf{b}^1) \leftarrow \left(\frac{1}{\gamma_1}, \widehat{\mathbf{W}}^1, \widehat{\mathbf{b}}^1\right)$, then the marginal distribution of each hidden unit $i$ becomes

$$P\left(h_i^1 = 1 \mid \mathbf{x}, \mathbf{W}^1, \mathbf{b}^1\right) = min\left\{\alpha_1 f\left(\widehat{\mathbf{W}}_i^1 \mathbf{x} + \widehat{b}_i^1\right), 1\right\} = \frac{1}{\gamma_1} f\left(\widehat{\mathbf{W}}_i^1 \mathbf{x} + \widehat{b}_i^1\right), \ \forall i, \mathbf{x} \in D. \quad (10)$$

---

[2]We do not use the extra SVHN dataset for training.

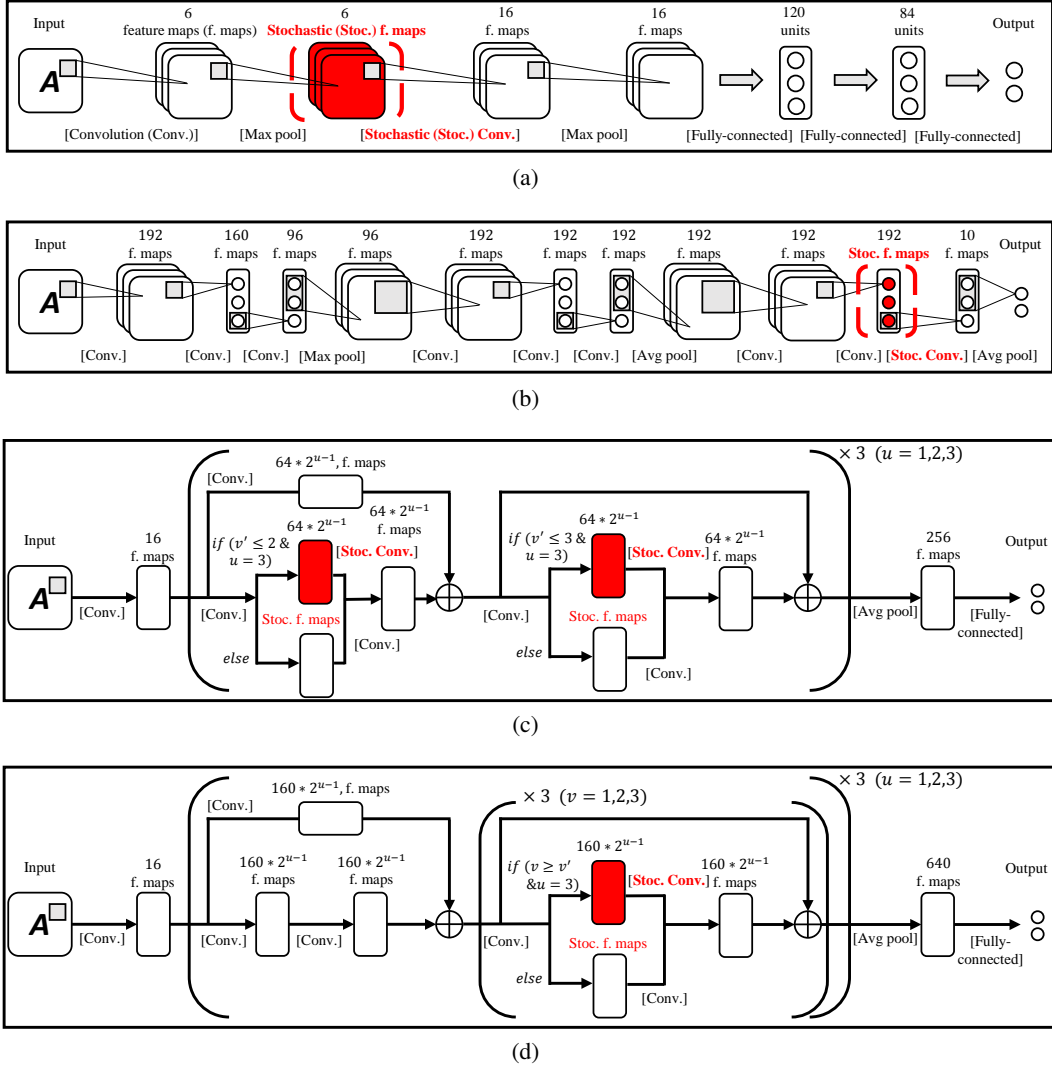

Figure 5: The overall structures of (a) Lenet-5, (b) NIN, (c) WRN with 16 layers, and (d) WRN with 28 layers. The red feature maps correspond to the stochastic ones. In case of WRN, we introduce one ($v' = 3$) and two ($v' = 2$) stochastic feature maps.

Next consider the second hidden layer. From Taylor's theorem, there exists a value $z$ between $0$ and $x$ such that $s(x) = s(0) + s'(0)x + R(x)$, where $R(x) = \frac{s''(z)x^2}{2!}$. Since we consider a binary random vector, i.e., $\mathbf{h}^1 \in \{0, 1\}^{N^1}$, one can write

$$\mathbb{E}_{P(\mathbf{h}^1|\mathbf{x})} \left[ s\left(\beta_j\left(\mathbf{h}^1\right)\right)\right] = \sum_{\mathbf{h}^1} \left(s\left(0\right) + s'\left(0\right)\beta_j\left(\mathbf{h}^1\right) + R\left(\beta_j\left(\mathbf{h}^1\right)\right)\right) P\left(\mathbf{h}^1 \mid \mathbf{x}\right)$$

$$= s\left(0\right) + s'\left(0\right)\left(\sum_i W_{ij}^2 P(h_i^1 = 1 \mid \mathbf{x}) + b_j^2\right) + \mathbb{E}_{P(\mathbf{h}^1|\mathbf{x})}\left[R(\beta_j(\mathbf{h}^1))\right],$$

where $\beta_j\left(\mathbf{h}^1\right) := \mathbf{W}_j^2\mathbf{h}^1 + b_j^2$ is the incoming signal. From (6) and (10), for every hidden unit $j$, it follows that

$$h_j^2\left(\mathbf{x}; \mathbf{W}^2, \mathbf{b}^2\right) = f\left(\alpha_2\left(s'(0)\left(\frac{1}{\gamma_1}\sum_i W_{ij}^2 \widehat{h}_i^1\left(\mathbf{x}\right) + b_j^2\right) + \mathbb{E}_{P(\mathbf{h}^1|\mathbf{x})}\left[R\left(\beta_j\left(\mathbf{h}^1\right)\right)\right]\right)\right).$$

Since we assume that $|f'(x)| \leq 1$, the following inequality holds:

$$\left| h_j^2(\mathbf{x}; \mathbf{W}^2, \mathbf{b}^2) - f\left( \alpha_2 s'(0) \left( \frac{1}{\gamma_1} \sum_i W_{ij}^2 \widehat{h}_i^1(\mathbf{x}) + b_j^2 \right) \right) \right| \leq \left| \alpha_2 \mathbb{E}_{P(\mathbf{h}^1 | \mathbf{x})} \left[ R(\beta_j(\mathbf{h}^1)) \right] \right|$$

$$\leq \frac{\alpha_2}{2} \mathbb{E}_{P(\mathbf{h}^1 | \mathbf{x})} \left[ \left( \mathbf{W}_j^2 \mathbf{h}^1 + b_j^2 \right)^2 \right],$$

where we use $|s''(z)| < 1$ for the last inequality. Therefore, it follows that

$$\left| h_j^2 \left( \mathbf{x}; \mathbf{W}^2, \mathbf{b}^2 \right) - \widehat{h}_j^2 \left( \mathbf{x}; \widehat{\mathbf{W}}^2, \widehat{\mathbf{b}}^2 \right) \right| \leq \frac{\gamma_1 \left( \sum_i \left| \widehat{W}_{ij}^2 \right| + \widehat{b}_j^2 \gamma_1^{-1} \right)^2}{2 s'(0) \gamma_2}, \quad \forall j,$$

since we set $\left( \alpha_2, \mathbf{W}^2, \mathbf{b}^2 \right) \leftarrow \left( \frac{\gamma_2 \gamma_1}{s'(0)}, \frac{\widehat{\mathbf{W}}^2}{\gamma_2}, \frac{\gamma_1^{-1}}{\gamma_2} \widehat{\mathbf{b}}^2 \right)$. This completes the proof of Theorem 1.

## D.2 PROOF OF THEOREM 2

For the proof of Theorem 2, we first state the two key lemmas on error propagation in Simplified-SFNN.

**Lemma 4** *Assume that there exists some positive constant $B$ such that*

$$\left| h_i^{\ell-1}(\mathbf{x}) - \widehat{h}_i^{\ell-1}(\mathbf{x}) \right| \leq B, \quad \forall i, \mathbf{x} \in D,$$

*and the $\ell$-th hidden layer of NCSFNN is standard deterministic layer as defined in (7). Given parameters $\{\widehat{\mathbf{W}}^\ell, \widehat{\mathbf{b}}^\ell\}$ of DNN, choose same ones for NCSFNN. Then, the following inequality holds:*

$$\left| h_j^\ell(\mathbf{x}) - \widehat{h}_j^\ell(\mathbf{x}) \right| \leq B N^{\ell-1} \widehat{W}_{\max}^\ell, \quad \forall j, \mathbf{x} \in D.$$

*where $\widehat{W}_{\max}^\ell = \max\limits_{ij} \left| \widehat{W}_{ij}^\ell \right|$.*

**Proof.** See Appendix D.3. □

**Lemma 5** *Assume that there exists some positive constant $B$ such that*

$$\left| h_i^{\ell-1}(\mathbf{x}) - \widehat{h}_i^{\ell-1}(\mathbf{x}) \right| \leq B, \quad \forall i, \mathbf{x} \in D,$$

*and the $\ell$-th hidden layer of simplified-SFNN is stochastic layer. Given parameters $\{\widehat{\mathbf{W}}^\ell, \widehat{\mathbf{W}}^{\ell+1}, \widehat{\mathbf{b}}^\ell, \widehat{\mathbf{b}}^{\ell+1}\}$ of DNN, choose those of Simplified-SFNN as follows:*

$$\alpha_\ell \leftarrow \frac{1}{\gamma_\ell}, \quad \left( \alpha_{\ell+1}, \mathbf{W}^{\ell+1}, \mathbf{b}^{\ell+1} \right) \leftarrow \left( \frac{\gamma_\ell \gamma_{\ell+1}}{s'(0)}, \frac{\widehat{\mathbf{W}}^{\ell+1}}{\gamma_{\ell+1}}, \frac{\widehat{\mathbf{b}}^{\ell+1}}{\gamma_\ell \gamma_{\ell+1}} \right),$$

*where $\gamma_\ell = \max\limits_{j, \mathbf{x} \in D} \left| f\left( \widehat{\mathbf{W}}_j^\ell \mathbf{h}^{\ell-1}(\mathbf{x}) + \widehat{b}_j^\ell \right) \right|$ and $\gamma_{\ell+1}$ is any positive constant. Then, it follows that*

$$\left| h_k^{\ell+1}(\mathbf{x}) - \widehat{h}_k^{\ell+1}(\mathbf{x}) \right| \leq B N^{\ell-1} N^\ell \widehat{W}_{\max}^\ell \widehat{W}_{\max}^{\ell+1} + \left| \frac{\gamma_\ell \left( N^\ell \widehat{W}_{\max}^{\ell+1} + \widehat{b}_{\max}^{\ell+1} \gamma_\ell^{-1} \right)^2}{2 s'(0) \gamma_{\ell+1}} \right|, \quad \forall k, \mathbf{x} \in D,$$

*where $\widehat{b}_{\max}^\ell = \max\limits_j \left| \widehat{b}_j^\ell \right|$ and $\widehat{W}_{\max}^\ell = \max\limits_{ij} \left| \widehat{W}_{ij}^\ell \right|$.*

**Proof.** See Appendix D.4. □

Assume that $\ell$-th layer is first stochastic hidden layer in Simplified-SFNN. Then, from Theorem 1, we have

$$\left| h_j^{\ell+1}(\mathbf{x}) - \widehat{h}_j^{\ell+1}(\mathbf{x}) \right| \leq \left| \frac{\gamma_\ell \left( N^\ell \widehat{W}_{\max}^{\ell+1} + \widehat{b}_{\max}^{\ell+1} \gamma_\ell^{-1} \right)^2}{2 s'(0) \gamma_{\ell+1}} \right|, \quad \forall j, \mathbf{x} \in D. \tag{11}$$

According to Lemma 4 and 5, the final error generated by the right hand side of (11) is bounded by

$$\frac{\tau_\ell \gamma_\ell \left( N^\ell \widehat{W}_{\max}^{\ell+1} + \widehat{b}_{\max}^{\ell+1} \gamma_\ell^{-1} \right)^2}{2 s'\left(0\right) \gamma_{\ell+1}}, \tag{12}$$

where $\tau_\ell = \prod_{\ell'=l+2}^{L} \left( N^{\ell'-1} \widehat{W}_{\max}^{\ell'} \right)$. One can note that every error generated by each stochastic layer is bounded by (12). Therefore, it follows that

$$\left| h_j^L\left(\mathbf{x}\right) - \widehat{h}_j^L\left(\mathbf{x}\right) \right| \leq \sum_{\ell:\text{stochastic hidden layer}} \left( \frac{\tau_\ell \gamma_\ell \left( N^\ell \widehat{W}_{\max}^{\ell+1} + \widehat{b}_{\max}^{\ell+1} \gamma_\ell^{-1} \right)^2}{2 s'\left(0\right) \gamma_{\ell+1}} \right), \quad \forall j, \mathbf{x} \in D.$$

From above inequality, we can conclude that

$$\lim_{\substack{\gamma_{\ell+1}\to\infty \\ \forall \text{ stochastic hidden layer } \ell}} \left| h_j^L\left(\mathbf{x}\right) - \widehat{h}_j^L\left(\mathbf{x}\right) \right| = 0, \quad \forall j, \mathbf{x} \in D.$$

This completes the proof of Theorem 2.

### D.3 PROOF OF LEMMA 4

From assumption, there exists some constant $\epsilon_i$ such that $|\epsilon_i| < B$ and

$$h_i^{\ell-1}\left(\mathbf{x}\right) = \widehat{h}_i^{\ell-1}\left(\mathbf{x}\right) + \epsilon_i, \quad \forall i, \mathbf{x}.$$

By definition of standard deterministic layer, it follows that

$$h_j^\ell\left(\mathbf{x}\right) = f\left( \sum_i \widehat{W}_{ij}^\ell h_i^{\ell-1}\left(\mathbf{x}\right) + \widehat{b}_j^{\ell-1} \right) = f\left( \sum_i \widehat{W}_{ij}^\ell \widehat{h}_i^{\ell-1}\left(\mathbf{x}\right) + \sum_i \widehat{W}_{ij}^\ell \epsilon_i + \widehat{b}_j^\ell \right).$$

Since we assume that $|f'(x)| \leq 1$, one can conclude that

$$\left| h_j^\ell\left(\mathbf{x}\right) - f\left( \sum_i \widehat{W}_{ij}^\ell \widehat{h}_i^{\ell-1}\left(\mathbf{x}\right) + \widehat{b}_j^\ell \right) \right| \leq \left| \sum_i \widehat{W}_{ij}^\ell \epsilon_i \right| \leq B \left| \sum_i \widehat{W}_{ij}^\ell \right| \leq B N^{\ell-1} \widehat{W}_{\max}^\ell.$$

This completes the proof of Lemma 4.

### D.4 PROOF OF LEMMA 5

From assumption, there exists some constant $\epsilon_i^{\ell-1}$ such that $\left| \epsilon_i^{\ell-1} \right| < B$ and

$$h_i^{\ell-1}\left(\mathbf{x}\right) = \widehat{h}_i^{\ell-1}\left(\mathbf{x}\right) + \epsilon_i^{\ell-1}, \quad \forall i, \mathbf{x}. \tag{13}$$

Let $\gamma_\ell = \max_{j, \mathbf{x} \in D} \left| f\left( \widehat{\mathbf{W}}_j^\ell \mathbf{h}^{\ell-1}(\mathbf{x}) + \widehat{b}_j^\ell \right) \right|$ be the maximum value of hidden units. If we initialize the parameters $\left( \alpha_\ell, \mathbf{W}^\ell, \mathbf{b}^\ell \right) \leftarrow \left( \frac{1}{\gamma_\ell}, \widehat{\mathbf{W}}^\ell, \widehat{\mathbf{b}}^\ell \right)$, then the marginal distribution becomes

$$P\left( h_j^\ell = 1 \mid \mathbf{x}, \mathbf{W}^\ell, \mathbf{b}^\ell \right) = \min\left\{ \alpha_\ell f\left( \widehat{\mathbf{W}}_j^\ell \mathbf{h}^{\ell-1}\left(\mathbf{x}\right) + \widehat{b}_j^\ell \right), 1 \right\} = \frac{1}{\gamma_\ell} f\left( \widehat{\mathbf{W}}_j^\ell \mathbf{h}^{\ell-1}\left(\mathbf{x}\right) + \widehat{b}_j^\ell \right), \; \forall j, \mathbf{x}.$$

From (13), it follows that

$$P\left( h_j^\ell = 1 \mid \mathbf{x}, \mathbf{W}^\ell, \mathbf{b}^\ell \right) = \frac{1}{\gamma_\ell} f\left( \widehat{\mathbf{W}}_j^\ell \widehat{\mathbf{h}}^{\ell-1}\left(\mathbf{x}\right) + \sum_i \widehat{W}_{ij}^\ell \epsilon_i^{\ell-1} + \widehat{b}_j^\ell \right), \quad \forall j, \mathbf{x}.$$

Similar to Lemma 4, there exists some constant $\epsilon_j^\ell$ such that $\left| \epsilon_j^\ell \right| < B N^{\ell-1} \widehat{W}_{\max}^\ell$ and

$$P\left( h_j^\ell = 1 \mid \mathbf{x}, \mathbf{W}^\ell, \mathbf{b}^\ell \right) = \frac{1}{\gamma_\ell} \left( \widehat{h}_j^\ell\left(\mathbf{x}\right) + \epsilon_j^\ell \right), \quad \forall j, \mathbf{x}. \tag{14}$$

Next, consider the upper hidden layer of stochastic layer. From Taylor's theorem, there exists a value $z$ between $0$ and $t$ such that $s(x) = s(0) + s'(0)x + R(x)$, where $R(x) = \frac{s''(z)x^2}{2!}$. Since we consider a binary random vector, i.e., $\mathbf{h}^\ell \in \{0, 1\}^{N^\ell}$, one can write

$$\mathbb{E}_{P(\mathbf{h}^\ell|\mathbf{x})}[s(\beta_k(\mathbf{h}^\ell))] = \sum_{\mathbf{h}^\ell} \left( s(0) + s'(0)\beta_k(\mathbf{h}^\ell) + R\left(\beta_k(\mathbf{h}^\ell)\right) \right) P(\mathbf{h}^\ell \mid \mathbf{x})$$

$$= s(0) + s'(0) \left( \sum_j W_{jk}^{\ell+1} P(h_j^\ell = 1 \mid \mathbf{x}) + b_k^{\ell+1} \right) + \sum_{\mathbf{h}^\ell} R(\beta_k(\mathbf{h}^\ell)) P(\mathbf{h}^\ell \mid \mathbf{x}),$$

where $\beta_k(\mathbf{h}^\ell) = \mathbf{W}_k^{\ell+1}\mathbf{h}^\ell + b_k^{\ell+1}$ is the incoming signal. From (14) and above equation, for every hidden unit $k$, we have

$$h_k^{\ell+1}(\mathbf{x}; \mathbf{W}^{\ell+1}, \mathbf{b}^{\ell+1})$$

$$= f\left( \alpha_{\ell+1} \left( s'(0) \left( \frac{1}{\gamma_\ell} \left( \sum_j W_{jk}^{\ell+1} \widehat{h}_j^\ell(\mathbf{x}) + \sum_j W_{jk}^{\ell+1} \epsilon_j^\ell \right) + b_k^{\ell+1} \right) + \mathbb{E}_{P(\mathbf{h}^\ell|\mathbf{x})} \left[ R(\beta_k(\mathbf{h}^\ell)) \right] \right) \right).$$

Since we assume that $|f'(x)| < 1$, the following inequality holds:

$$\left| h_k^{\ell+1}(\mathbf{x}; \mathbf{W}^{\ell+1}, \mathbf{b}^{\ell+1}) - f\left( \alpha_{\ell+1} s'(0) \left( \frac{1}{\gamma_\ell} \sum_j W_{ij}^{\ell+1} \widehat{h}_j^\ell(\mathbf{x}) + b_j^{\ell+1} \right) \right) \right|$$

$$\leq \left| \frac{\alpha_{\ell+1} s'(0)}{\gamma_\ell} \sum_j W_{jk}^{\ell+1} \epsilon_j^\ell + \alpha_{\ell+1} \mathbb{E}_{P(\mathbf{h}^\ell|\mathbf{x})} \left[ R(\beta_k(\mathbf{h}^\ell)) \right] \right|$$

$$\leq \left| \frac{\alpha_{\ell+1} s'(0)}{\gamma_\ell} \sum_j W_{jk}^{\ell+1} \epsilon_j^\ell \right| + \left| \frac{\alpha_{\ell+1}}{2} \mathbb{E}_{P(\mathbf{h}^\ell|\mathbf{x})} \left[ \left( \mathbf{W}_k^{\ell+1}\mathbf{h}^\ell + b_k^{\ell+1} \right)^2 \right] \right|, \tag{15}$$

where we use $|s''(z)| < 1$ for the last inequality. Therefore, it follows that

$$\left| h_k^{\ell+1}(\mathbf{x}) - \widehat{h}_k^{\ell+1}(\mathbf{x}) \right| \leq B N^{\ell-1} N^\ell \widehat{W}_{\max}^\ell \widehat{W}_{\max}^{\ell+1} + \left| \frac{\gamma_\ell \left( N^\ell \widehat{W}_{\max}^{\ell+1} + \widehat{b}_{\max}^{\ell+1} \gamma_\ell^{-1} \right)^2}{2 s'(0) \gamma_{\ell+1}} \right|,$$

since we set $\left( \alpha_{\ell+1}, \mathbf{W}^{\ell+1}, \mathbf{b}^{\ell+1} \right) \leftarrow \left( \frac{\gamma_{\ell+1}\gamma_\ell}{s'(0)}, \frac{\widehat{\mathbf{w}}^{\ell+1}}{\gamma_{\ell+1}}, \frac{\gamma_\ell^{-1}\widehat{\mathbf{b}}^{\ell+1}}{\gamma_{\ell+1}} \right)$. This completes the proof of Lemma 5.

