# Peer review of "Making Stochastic Neural Networks from Deterministic Ones"

_ICLR 2017 — rejected_

[Public Comment · Tara N Sainath · 07 Nov 2016]
**ICLR Paper Format**

Dear Authors,

Please resubmit your paper in the ICLR 2017 format with the margins to the correct spacing for your submission to be considered. Thank you!

[Reviewer Comment · AnonReviewer1 · 12 Dec 2016]
**Question about Table 1**

In Table 1, do the 4-layer SFNNs have one or two layers of stochastic units?  What about the 3-layer networks?  I suppose you could take the expectation in the output layer.

[Reviewer Comment · AnonReviewer1 · 12 Dec 2016]
**Citation format does not match the ICLR template**

In this paper, citations are appearing with the authors' first initials and last names, e.g. (Hinton, G. et al., 2012a) instead of the authors last names and no initials, e.g. (Hinton et al., 2012a).  I find the first initials to be very distracting.  Please reformat the paper to match the citation style of the ICLR 2017 template.

[Official Review · AnonReviewer1 · rating 5 · confidence 4 · 14 Dec 2016 (modified: 25 Jan 2017)]
**Promising MNIST classification results, but stronger baseline on CIFAR-10, CIFAR-100, or SVHN would have been nice**

Update: Because no revision of the paper has been provided by the authors, I am reducing my rating to "marginally below acceptance".

----------

This paper addresses the problem of training stochastic feedforward neural networks.  It proposes to transfer weights from a deterministic deep neural network trained using standard procedures (including techniques such as dropout and batch normalization) to a stochastic network having the same topology.  The initial mechanism described for performing the transfer involves a rescaling of unit inputs and layer weights, and appropriate specification of the stochastic latent units if the DNN used for pretraining employs ReLU nonlinearities.  Initial experiments on MNIST classification and a toy generative task with a multimodal target distribution show that the simple transfer process works well if the DNN used for pretraining uses sigmoid nonlinearities, but not if the pretraining DNN uses ReLUs.  To tackle this problem, the paper introduces the "simplified stochastic feedforward neural network," in which every stochastic layer is followed by a layer that takes an expectation over samples from its input, thus limiting the propagation of stochasticity in the network.  A modified process for transferring weights from a pretraining DNN to the simplified SFNN is described and justified.  The training process then occurs in three steps:  (1) pretrain a DNN, (2) transfer weights from the DNN to a simplified SFNN and continue training, and (3) optionally transfer the weights to a full SFNN and continue training or transfer them to a deterministic model (called DNN*) and continue training.  The third step can be skipped and the simplified SFNN may also be used directly as an inference model.  Experimental results on MNIST classification show that the use of simplified SFNN training can improve a deterministic DNN* model over a DNN baseline trained with batch normalization and dropout.  Experiments on two generative tasks (MNIST-half and the Toronto Faces Database) show that the proposed pretraining process improves test set negative log-likelihoods.  Finally, experiments on CIFAR-10, CIFAR-100, and SVHN with the LeNet-5, network-in-network, and wide residual network architectures show that use of a stochastic training step can improve performance of a deterministic (DNN*) model.

It is a bit confusing to refer to "multi-modal" tasks, when what is meant is "generative tasks with a multimodal target distribution" because "multi-modal" task can also refer to a learning task that crosses sensory modalities such as audio-visual speech recognition, text-based image retrieval, or image captioning.  I recommend that you use the more precise term ("generative tasks with a multimodal target distribution") early in the introduction and then say that you will refer to such tasks as "multi-modal tasks" in the rest of the paper for the sake of brevity.

The paper would be easier to read if "SFNN" were not used to refer to both the singular ("stochastic feedforward neural network") and plural ("stochastic feedforward neural networks") cases.  When the plural is meant, write "SFNNs".

In Table 1, why does the 3 hidden layer SFNN initialized from a ReLU DNN have so much worse of a test NLL than the 2 hidden layer SFNN initialized from a ReLU DNN?

The notation that uses superscripts to indicate layer indexes is confusing.  The reader naturally parses N² as "N squared" and not as "the number of units in the second layer."

When you transfer weights back from the simplified SFNN to the DNN* model, do you need to perform some sort of rescaling that undoes the operations in Equation (8) in the paper?

What does NCSFNN stand for in the supplementary material?

Pros
+ The proposed model is easy to implement and apply to other tasks.
+ The MNIST results showing that the stochastic model training can produce a deterministic model (called DNN* in the paper) that generalizes better than a DNN trained with batch normalization and dropout is quite exciting.

Cons
- For the reasons outlined above, the paper is at times a bit hard to follow.
- The results CIFAR-10, CIFAR-100, and SVHN would be more convincing if the baselines used dropout and batch normalization.  While this is shown on MINST, demonstration of a similar result on a more challenging task would strengthen the paper.

Minor issues

It has been believed that stochastic → It is believed that stochastic

underlying these successes is on the efficient training methods → underlying these successes is efficient training methods

necessary in order to model complex stochastic natures in many real-world tasks → necessary in to model the complex stochastic nature of many real-world tasks

structured prediction, image generation and memory networks : memory networks are models, not tasks.

Furthermore, it has been believed that SFNN → Furthermore, it is believed that SFNN

using backpropagation under the variational techniques and the reparameterization tricks  → using backpropagation with variational techniques and reparameterization tricks

There have been several efforts developing efficient training methods → There have been several efforts toward developing efficient training methods

However, training SFNN is still significantly slower than doing DNN → However, training a SFNN is still significantly slower than training a DNN

e.g., most prior works on this line have considered a → consequently most prior works in this area have considered a

Instead of training SFNN directly → Instead of training a SFNN directly

whether pre-trained parameters of DNN → whether pre-trained parameters from a DNN

with further fine-tuning of light cost → with further low-cost fine-tuning

recent advances in DNN on its design and training → recent advances in DNN design and training

it is rather believed that transferring parameters →  it is believed that transferring parameters

but the opposite direction is unlikely possible → but the opposite is unlikely

To address the issues, we propose → To address these issues, we propose

which intermediates between SFNN and DNN, → which is intermediate between SFNN and DNN,

in forward pass and computing gradients in backward pass → in the forward pass and computing gradients in the backward pass

in order to handle the issue in forward pass →  in order to handle the issue in the forward pass

Neal (1990) proposed a Gibbs sampling → Neal (1990) proposed Gibbs sampling

for making DNN and SFNN are equivalent → for making the DNN and SFNN equivalent

in the case when DNN uses the unbounded ReLU → in the case when the DNN uses the unbounded ReLU

are of ReLU-DNN type due to the gradient vanishing problem → are of the ReLU-DNN type because they mitigate the gradient vanishing problem

multiple modes in outupt space y → multiple modes in output space y

The only first hidden layer of DNN → Only the first hidden layer of the DNN

is replaced by stochastic one, → is replaced by a stochastic layer,

the former significantly outperforms for the latter for the → the former significantly outperforms the latter for the

simple parameter transformations from DNN to SFNN are not clear to work in general, → simple parameter transformations from DNN to SFNN do not clearly work in general,

is a special form of stochastic neural networks → is a special form of stochastic neural network

As like (3), the first layer is → As in (3), the first layer is

This connection naturally leads an efficient training procedure → This connection naturally leads to an efficient training procedure

[Reviewer Comment · AnonReviewer1 · 14 Dec 2016]
**How is simplified SFNN to DNN* transfer performed?**

When you transfer weights back from the simplified SFNN to the DNN* model, do you need to perform some sort of rescaling that undoes the operations in Equation (8) in the paper?

[Official Review · AnonReviewer3 · rating 6 · confidence 4 · 16 Dec 2016]
**interesting connection between DNN and simplified SFNN but its practical significance is unknown**

This paper builds connections between DNN, simplified stochastic neural network (SFNN) and SFNN and proposes to use DNN as the initialization model for simplified SFNN. The authors evaluated their model on several small tasks with positive results.

The connection between different models is interesting. I think the connection between sigmoid DNN and Simplified SFNN is the same as mean-field approximation that has been known for decades. However, the connection between ReLU DNN and simplified SFNN is novel.

My main concern is whether the proposed approach is useful when attacking real tasks with large training set. For tasks with small training set I can see that stochastic units would help generalize well.

[Official Review · AnonReviewer2 · rating 5 · confidence 5 · 24 Dec 2016]
**The connection between different models is interesting, except for Bayesian net which is superficial and need to discuss more; MNIST results are interesting but more tasks need to be explored.**

Strengths

- interesting to explore the connection between ReLU DNN and simplified SFNN
- small task (MNIST)  is used to demonstrate the usefulness of the proposed training methods experimentally
- the proposed, multi-stage training methods are simple to implement (despite lacking theoretical rigor)


Weaknesses

-no results are reported on real tasks with large training set

-not clear exploration on the scalability of the learning methods when training data becomes larger

-when the hidden layers become stochastic, the model shares uncertainty representation with deep Bayes networks or deep generative models (Deep Discriminative and Generative Models for Pattern Recognition , book chapter in “Pattern Recognition and Computer Vision”, November 2015, Download PDF). Such connections should be discussed, especially wrt the use of uncertainty representation to benefit pattern recognition (i.e. supervised learning via Bayes rule) and to benefit the use of domain knowledge such as “explaining away”.

-would like to see connections with variational autoencoder models and training, which is also stochastic with hidden layers

[Final Decision · Program Chairs · 06 Feb 2017]
**ICLR committee final decision**

No reviewer was willing to champion the paper and the authors did not adequately address reviewer comments in a revision. Recommend rejection.